# Improving the Energy Storage Performance of Barium Titanate-Based Ceramics through the Addition of ZnO-Bi$_2$O$_3$-SiO$_2$ Glass

**Peifeng Xiong** [1,†], **Man Xiao** [1,†], **Zhonghua Yao** [1], **Hanxing Liu** [1] and **Hua Hao** [1,2,*]

1   State Key Laboratory of Advanced Technology for Materials Synthesis and Processing, School of Material Science and Engineering, International School of Material Science and Engineering, Wuhan University of Technology, Wuhan 430070, China; 265515@whut.edu.cn (P.X.); sun0212@whut.edu.cn (M.X.); yaozhhua@whut.edu.cn (Z.Y.); lhxhp@whut.edu.cn (H.L.)

2   Foshan Xianhu Laboratory of the Advanced Energy Science and Technology Guangdong Laboratory, Xianhu Hydrogen Valley, Foshan 528200, China

\*   Correspondence: haohua@whut.edu.cn

†   These authors contributed equally to this work.

**Abstract:** Lead-free ceramics with excellent energy storage performance are important for high-power energy storage devices. In this study, 0.9BaTiO$_3$-0.1Bi(Mg$_{2/3}$Nb$_{1/3}$)O$_3$ (BT-BMN) ceramics with $x$ wt% ZnO-Bi$_2$O$_3$-SiO$_2$ (*ZBS*) ($x$ = 2, 4, 6, 8, 10) glass additives were fabricated using the solid-state reaction method. X-ray diffraction (XRD) analysis revealed that the *ZBS* glass-added ceramics exhibited a perovskite structure, with the maximum relative density achieved at $x$ = 6. The average grain size reduced obviously as the glass additive wt% increased. Also, the dielectric constant decreased and the breakdown strength increased with increases in the glass additives. The optimal energy storage density of 1.39 J/cm$^3$ with an energy storage efficiency of 78.3% was obtained at $x$ = 6 due to high maximum polarization and enhanced breakdown strength. The results demonstrate that this material is a potential candidate for high-pulse-power energy storage devices.

**Keywords:** glass additives; BaTiO$_3$; energy storage properties; dielectric properties

## 1. Introduction

Barium titanate-based (BaTiO$_3$-based) ceramics have been actively studied over the past few decades as dielectric materials in energy storage applications due to their high power density, fast charge/discharge rate, and high stability [1–5]. To design a proper energy storage dielectric material, high maximum polarization ($P_{max}$), low remanent polarization ($P_r$), and high breakdown strength (BDS) should be satisfied at the same time. Numerous investigations have focused on improving the energy storage performance of BaTiO$_3$-based ceramics.

Relaxor ferroelectrics exhibit high $P_{max}$, low $P_r$, and moderate BDS, making them promising candidate materials for energy storage applications [6–9]. In recent years, BaTiO$_3$-BiMeO$_3$ [Me = Sc$^{3+}$, (Mg$_{1/2}$Ti$_{1/2}$)$^{3+}$, (Ni$_{2/3}$Nb$_{1/3}$)$^{3+}$, etc.] as classical relaxor ferroelectrics have attracted extensive attractions since the introduction of Bi-based perovskite can improve grain density and obtain high $P_{max}$ [10–13]. Of particular interest is that the 0.9BaTiO$_3$-0.1Bi(Mg$_{2/3}$Nb$_{1/3}$)O$_3$ ceramic was reported to achieve a maximum polarization of 16.57 μC/cm$^2$ and a recoverable energy storage density of 1.13 J/cm$^3$ at 143.5 kV/cm [14]; however, its relatively low BDS limits its potential applications.

The BDS of ceramics is substantially influenced by several factors, such as the grain size, grain density, porosity, second phase, and interfacial polarization [15–19]. Glass additives can decrease the sintering temperature and refine the grain size of ceramics. S. Yoon et al. [20] investigated TiO$_2$ ceramics with added ZnO-Bi$_2$O$_3$-SiO$_2$ (*ZBS*) glass and found that glass additives can lower the sintering temperature and enhance the density

of ceramics, and improve their dielectric properties. The addition of ZnO can promote moderate grain growth and improve the uniformity of the microstructure of ceramics. Dong et al. [21] utilized this method to increase the energy storage density of $Ba_{0.3}Sr_{0.7}TiO_3$ ceramics. $Bi_2O_3$, when serving as a sintering aid, can significantly reduce the sintering temperature due to the effect of liquid-phase sintering [22]. $SiO_2$ can inhibit grain growth, which is beneficial for enhanced ceramic density. Lee et al. [23] incorporated $SiO_2$ into $BaTiO_3$-based ceramics, resulting in ceramic materials with a density exceeding 95% and improved dielectric properties. In summary, the introduction of $ZnO$-$Bi_2O_3$-$SiO_2$ (*ZBS*) glass can improve the microstructure and sintering properties of ceramics, consequently enhancing their energy storage performance.

In the present work, to improve the energy storage performance of barium titanate-based ceramics, *ZBS* glass samples to be used as additives for $0.9BaTiO_3$-$0.1Bi(Mg_{2/3}Nb_{1/3})O_3$ (referred to as BT-BMN) ceramics were prepared. The effects of these glass additives on the microstructures, dielectric properties, breakdown strengths, and energy storage properties of the ceramics were systematically investigated.

## 2. Materials and Methods

### 2.1. Glass Additive Fabrication

The compositions of the glass were $40ZnO$-$35Bi_2O_3$-$25SiO_2$ (mol%) and $50ZnO$-$30Bi_2O_3$-$20SiO_2$ (mol%) (referred to as *ZBS1* and *ZBS2*). Calculated amounts of these chemicals were ball-milled for 4 h and then melted in a corundum crucible at 1450 °C for 2 h. The melt was then quenched in water to obtain glass powder. Subsequently, the powders were ball-milled for 1 h and sieved through a 60-mesh screen to produce powders with fine particles.

### 2.2. Sample Preparation

The BT-BMN powder was fabricated using the solid-state reaction method. High-purity powders $BaCO_3$ (>99%), $TiO_2$ (>99%), $Bi_2O_3$ (>99%), MgO (>98%), and $Nb_2O_5$ (>99%) were used as the raw material according to the stoichiometry of BT-BMN. The weighed raw materials were ball-milled in alcohol for 6 h. After drying, the powders were calcined at 1000 °C for 4 h in sealed alumina crucibles.

The BT-BMN powders were mixed with the glass additives by ball milling for 6 h according to the following weight ratio: $(100 - x)$ wt% BT-BMN + $x$ wt% ($x$ = 2, 4, 6, 8, 10) glass (depicted as *Z1*, *Z2*, *Z3*, *Z4*, and *Z5*, respectively). The mixed powders were pressed into pellets with the dimensions of 10 mm in diameter and 1 mm in thickness under a pressure of 200 MPa. The pellets were sintered at different temperatures from 1080 to 1150 °C for 2 h. The sintered pellets were polished down to a thickness of 0.3 mm.

### 2.3. Characterization

The bulk densities of the samples were measured using the Archimedes method. The X-ray diffraction (XRD) patterns of the additives and glass ceramics were analyzed using a PANalytical X'Pert-PRO diffractometer (Eindhoven, The Netherlands) at 40 mA and 40 kV. The microstructures of the samples were observed using a JSM-6700F Scanning Election Microscope (SEM) (JEOL, Tokyo, Japan). The samples were coated using a Ag electrode with a diameter of 2 mm for the dielectric and ferroelectric measurements. The dielectric data were collected using a precision LCR meter (4284A, HP, USA) from −180 °C to 200 °C with the measuring frequency of 1 kHz to 1 MHz. The polarization–electric field (*P–E*) hysteresis loops were assessed using the TF Analyzer 2000 (aixACCT, Aachen, Nordrhein-Westfalen, Germany) ferroelectric test system at room temperature and a frequency of 10 Hz.

## 3. Results and Discussion

### 3.1. Phase Structure

Figure 1a,b show the XRD patterns of glass additives *ZBS1* and *ZBS2*, respectively. It can be observed that the XRD results for *ZBS1* exhibit no significant peaks, indicating no appearance of crystallization and confirming the completion of glass fabrication, while *ZBS2*

shows crystallization. As the ZnO content in the glass system increases, the crystallization temperature of the glass tends to decrease. In addition, ZnO can facilitate grain nucleation and make glass susceptible to crystallization. Therefore, for the subsequent research, the *ZBS1* glass was chosen for addition into the BT-BMN ceramics. The properties of this *ZBS1* glass are shown in Table 1.

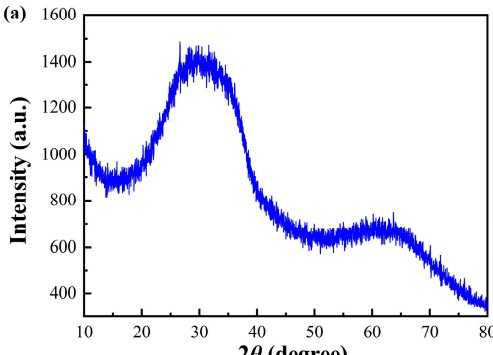 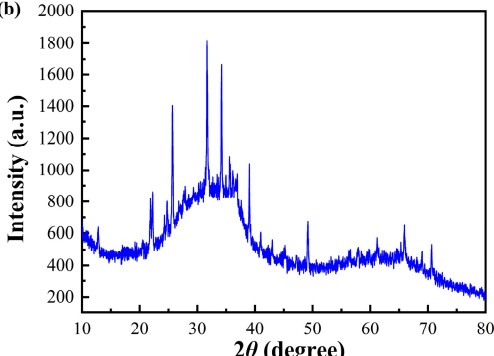

**Figure 1.** XRD patterns of (**a**) *ZBS1* and (**b**) *ZBS2* glass additives.

**Table 1.** Properties of the *ZBS1* glass.

| Density (g/cm$^3$) | Softening Point (°C) | Dielectric Constant $\varepsilon_r$ | Dielectric Loss $tan\ \sigma$ | Breakdown Strength (kV/cm) |
|---|---|---|---|---|
| 3.001 | ~700 | ~6.3 | 0.001 | 358 |

The effect of sintering temperature on bulk density in the glass-added BT-BMN ceramics is shown in Figure 2. It can be observed that the bulk densities of the *Z1* and *Z2* samples exhibit an initial increase followed by a decrease with increases in the sintering temperature, while the *Z3*, *Z4*, and *Z5* samples exhibit an overall decreasing trend in their bulk densities with the increasing sintering temperature. The sintering temperature required to achieve maximum bulk density is 1100 °C for the *Z1* and *Z2* samples, whereas this decreases to 1080 °C for the *Z3*, *Z4*, and *Z5* samples. With increases in the glass additive content, the optimal sintering temperature shows a decreasing trend. Due to the low melting point of *ZBS*, when the temperature reaches 700 °C, the *ZBS* glass begins to soften and forms a liquid phase as the sintering temperature rises. The formation of a liquid phase in the sintering process can reduce the sintering activation energy, which can promote material transformation during sintering and the densification of ceramics; this consequently improves the density and refines the grain size, resulting in enhanced energy storage performance [24,25].

Figure 3 shows the XRD patterns of BT-BMN ceramics with various amounts of *ZBS* glass. The diffraction peaks of each sample exhibit no shifting or splitting, indicating that all of the samples exhibit a perovskite structure without a secondary phase. The XRD results after refinement with the program X'Pert HighScore Plus (Version 3.0.5, PANalytical, the Netherlands)and the crystal structure details are shown in Table 2. The lattice parameter of each component varies slightly with the increasing content of *ZBS1* glass additives, indicating that the crystal structure of the main phase is unchanged. The changes in crystal lattice parameters can be attributed to substitution effects. All samples exhibit a perovskite ABO$_3$ structure, where the B-site comprises Ti$^{4+}$, Mg$^{2+}$, and Nb$^{5+}$. The ionic radius of Ti$^{4+}$ is 0.604 Å, while the equivalent ionic radii of Mg$^{2+}$ and Nb$^{5+}$ are 0.693 Å. During the sintering process, the formation of a liquid phase involving *ZBS* glass participates in the mass transfer process. In this context, Zn$^{2+}$ ions in the *ZBS* glass may be substituted into the lattice due to their proximity to the B-site ions (the ionic radius for Zn$^{2+}$ is 0.74 Å), leading to changes in the lattice parameters [26,27]. The relative density of the ceramics with different levels of *ZBS1* glass content exhibits a trend of initially increasing and then

decreasing. When the glass content reaches 6% and 8%, the relative density exceeds 95%, demonstrating a relatively dense structure of the ceramic samples.

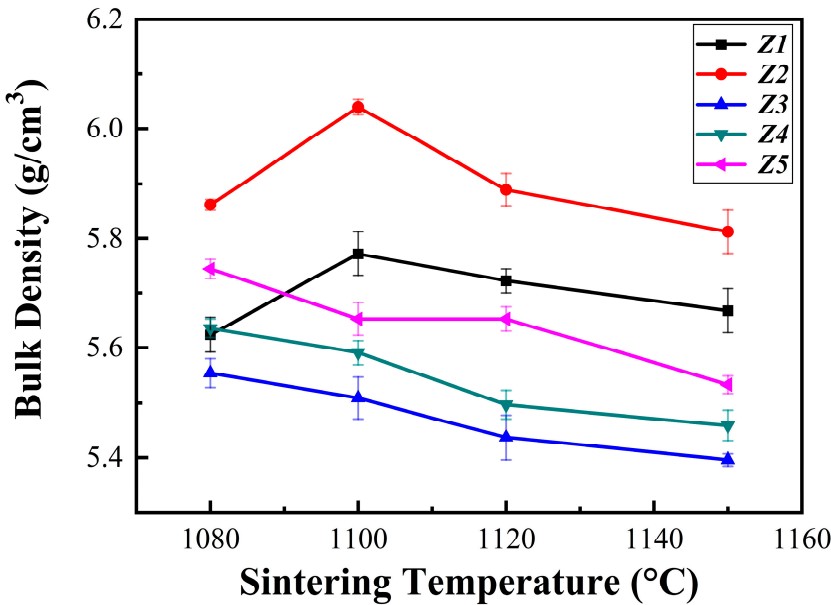

**Figure 2.** Bulk density as a function of sintering temperature for the glass-added BT-BMN ceramics.

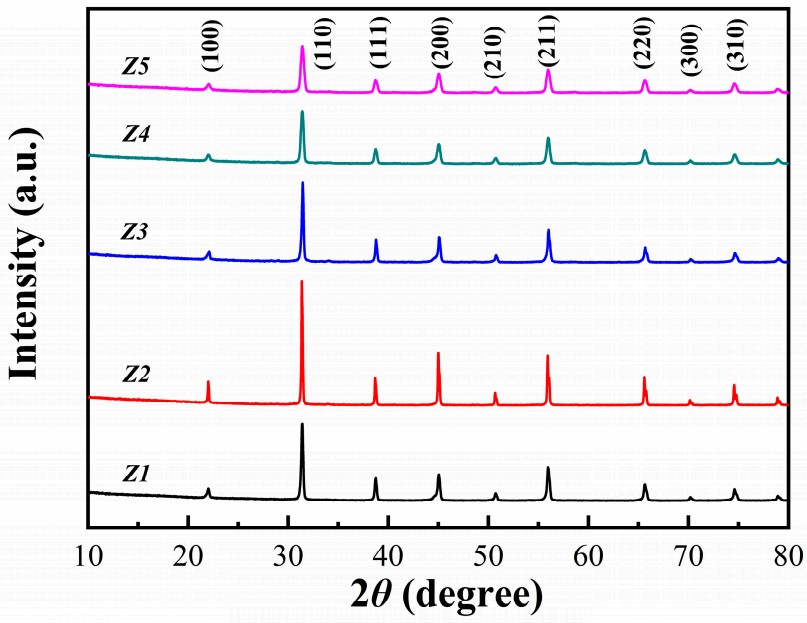

**Figure 3.** XRD patterns of the glass-added BT-BMN ceramics.

**Table 2.** Lattice parameters and relative density of the glass-added BT-BMN ceramics.

| Sample | Structure | Lattice Parameter (Å) | Volume (Å$^3$) | Theoretical Density (g/cm$^3$) | Bulk Density (g/cm$^3$) | Relative Density (%) |
|---|---|---|---|---|---|---|
| Z1 | Cubic | 4.0153 | 64.74 | 6.40 | 5.77 | 90.13 |
| Z2 | Cubic | 4.0182 | 64.88 | 6.68 | 6.04 | 90.47 |
| Z3 | Cubic | 4.0349 | 65.20 | 5.64 | 5.55 | 97.55 |
| Z4 | Cubic | 4.0192 | 64.92 | 5.88 | 5.64 | 95.80 |
| Z5 | Cubic | 4.0175 | 64.81 | 6.13 | 5.74 | 94.67 |

### 3.2. Microstructure Analysis

The SEM micrographs of BT-BMN ceramics supplemented with different amounts of glass are shown in Figure 4. The SEM micrographs of all samples clearly show the presence of grains, indicating their good crystallinity. As the glass content increases, there is a significant reduction in the grain size of the ceramic samples, with the average grain size decreasing from 1.29 μm to 0.55 μm. The results of the grain size observation can correspond to the XRD patterns of the samples. With the increasing content of glass additives, the XRD patterns of the samples exhibit broader diffraction peaks. This suggests a reduction in the size of coherent scattering regions (CSRs), indicating smaller grain sizes, which aligns well with the findings from the SEM micrographs. During the sintering process, glass additives dissolve at the grain boundaries or interfaces of ceramics, forming a liquid phase. This liquid phase helps to reduce grain boundary energy, reducing the obstruction between the grains and facilitating grain movement and rearrangement, ultimately refining the grain size of the ceramics. Theoretically, a smaller grain size implies more grain boundaries within the material, which increases the resistance and enhances the breakdown strength. Ceramics with smaller grain sizes require relatively lower energy for polarization reversal, facilitating easier polarization reversal and resulting in a higher energy storage density. When the glass content reaches 10%, a marginal increase in the porosity of the ceramics is noted. This may be attributed to an excessive glass content, leading to an increase in the viscosity of the ceramics, and consequently impeding the expulsion of pores within the ceramics. Observing the results of the SEM tests, it is evident that the grains in the *Z3* sample exhibit a homogeneous structure with very few pores, corresponding to the relative density result in Table 2.

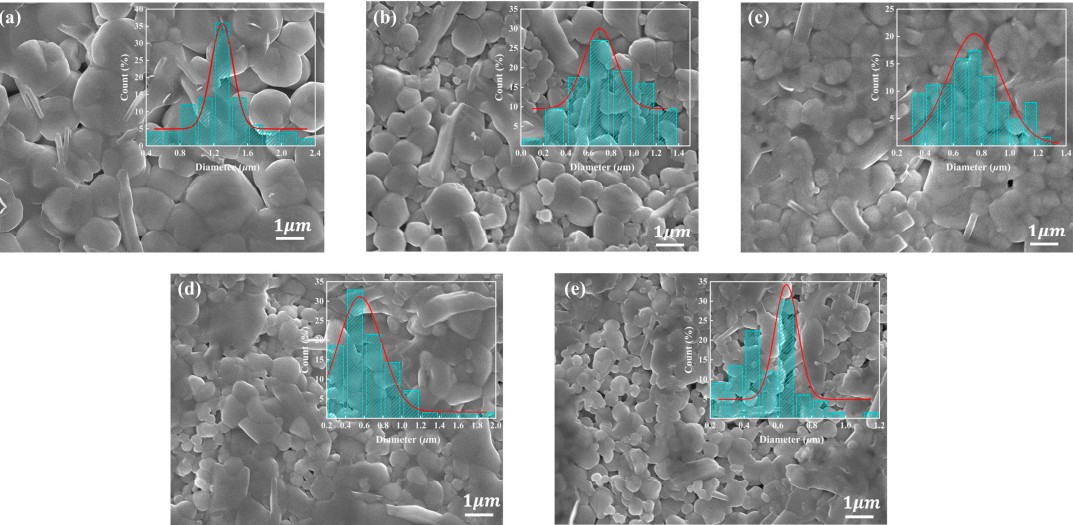

**Figure 4.** SEM micrographs of the glass-added BT-BMN ceramics: (**a**) *x* = 2, *Z1*; (**b**) *x* = 4, *Z2*; (**c**) *x* = 6, *Z3*; (**d**) *x* = 8, *Z4*; (**e**) *x* = 10, *Z5*. The insets show the grain size distributions in the ceramics.

### 3.3. Dielectric Behavior

Figure 5 displays the temperature dependence of the dielectric constant and dielectric loss of the glass-added BT-BMN ceramics as a function of the frequency. The dielectric constant and dielectric loss of the samples with varying glass additive contents exhibit distinctly broadened phase transition and frequency dispersion behavior, which indicates that the dielectric constant and dielectric loss can remain stable within a certain temperature range. The dielectric constant shows a decreasing trend with increases in the frequency, which indicates a relaxor-type behavior [28]. The maximum dielectric constants at 1 kHz decrease from 1020.8 to 624.3 as the glass content increases. In this glass–ceramic system, the dielectric constant of the *ZBS* glass is only 6.3, while that of the BT-BMN ceramic

is ~1200, according to Lichtenecker's equation [29] (Equation (1)), causing the dielectric constant to inevitably decrease:

$$log\varepsilon_r = \sum_i V_i log\varepsilon_{r_i} \tag{1}$$

where $\varepsilon_r$ is the relative dielectric constant of the glass–ceramic composite, $V_i$ is the relative volume fraction, and $\varepsilon_{r_i}$ is the relative dielectric constant of the components. The dielectric loss tangent for the samples shows an increasing trend with the testing frequency, caused by the ion jump relaxation at higher frequencies [30]. The dielectric loss shows minimal variation with the addition of glass, in the range of 0.005~0.007.

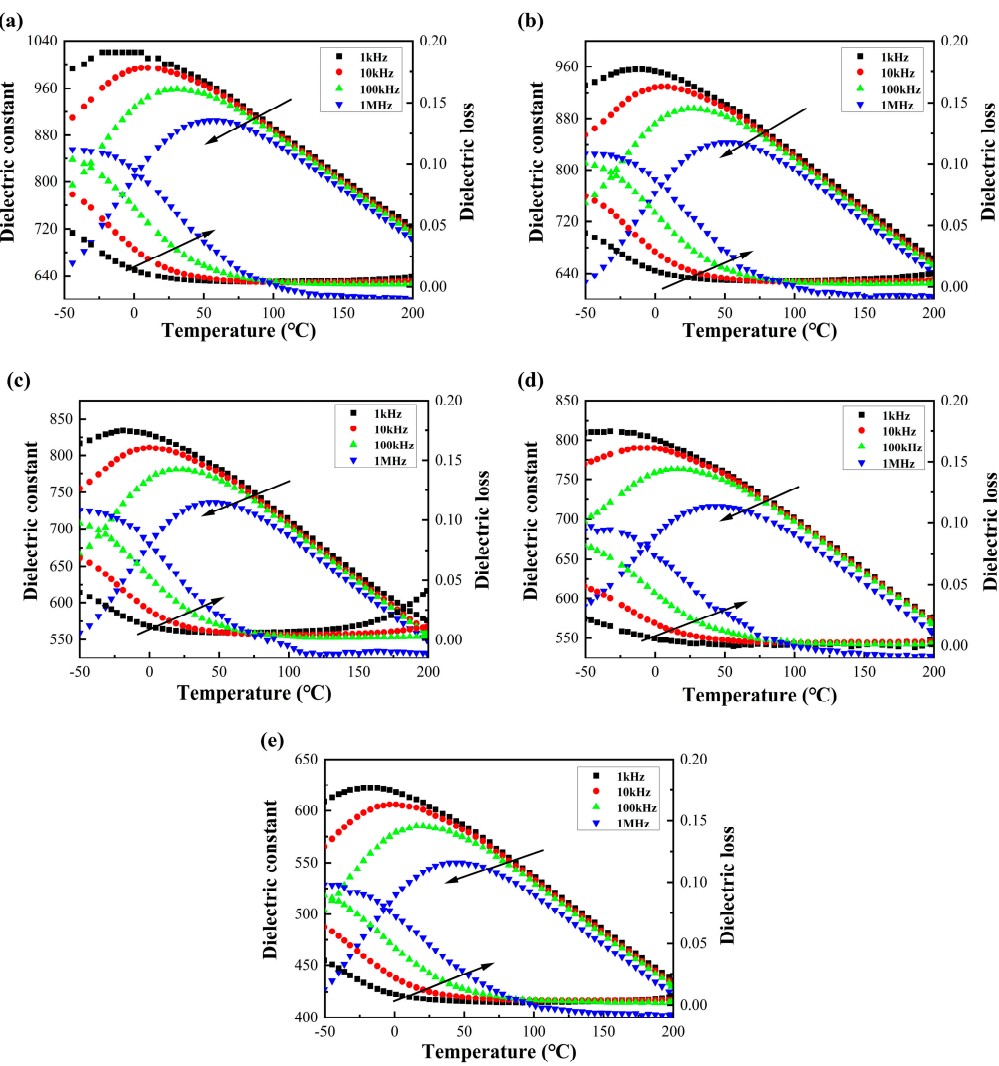

**Figure 5.** Temperature-dependent dielectric characteristics of the glass−added BT-BMN ceramics: (**a**) *x* = 2, *Z1*; (**b**) *x* = 4, *Z2*; (**c**) *x* = 6, *Z3*; (**d**) *x* = 8, *Z4*; (**e**) *x* = 10, *Z5*. The arrows in the figures represent the trends of the dielectric constant and dielectric loss as the frequency increases.

### *3.4. Ferroelectric Behavior and Energy Storage Performance*

The ferroelectric and energy storage behavior of the glass-added BT-BMN ceramics are investigated based on the polarization–electric field (*P–E*) hysteresis loops at room temperature and 1 Hz prior to their respective breakdown strength, as shown in Figure 6a. The total energy density (*W*) and recoverable energy density (*W_{rec}*) are calculated from the integral area of *P–E* loops based on the following equations:

$$W = \int_0^{P_{max}} E\,dP \tag{2}$$

$$W_{rec} = \int_{P_r}^{P_{max}} E\,dP \tag{3}$$

where $E$ is the applied electric field, $P_{max}$ is the maximum polarization, and $P_r$ is the remanent polarization. The energy storage efficiency $\eta$ is expressed as $\eta = W_{rec}/W$. The BDS of glass-added ceramics increases with the addition of *ZBS1* glass, as shown in Figure 6b. The microstructure of the materials, including porosity, grain density, and grain size, is an important influence factor for the BDS of ceramics. Microstructure analysis indicates that the enhancement in BDS is mainly due to the fact that the *ZBS*1 addition can refine the grains and make the grains distribute uniformly. Despite the lower relative density and higher porosity of *Z5* compared to *Z3* and *Z4*, the reduction in grain size contributes to increased grain boundaries, which increases the resistance and provides additional dislocation, leading to more efficient charge trapping and dissipation, and thereby enhancing breakdown strength. This dominant effect in enhancing the BDS of ceramics results in the highest BDS being observed in the *Z5* sample, reaching 228 kV/cm. The maximum $W$ and $W_{rec}$ (1.77 J/cm$^3$ and 1.39 J/cm$^3$, respectively) are obtained when $x = 6$, the same trend as the $P_{max}$, indicating that the energy density is mainly dominated by polarization. The $P_{max}$ of the *Z3* sample reaches 17.65 μC/cm$^2$, representing a 6.6% increase compared to the pure BT-BMN [14]. The *Z5* sample exhibits the highest BDS but its $P_{max}$ is low and $P_r$ is high, resulting in a relatively low energy storage density. Among all of the measured samples, *Z3* has the highest $W_{rec}$ at 1.39 J/cm$^3$, with an ideal $\eta$ of 78.3%, showing excellent energy storage performance. The increased energy density of these glass-added BT-BMN ceramics can be attributed to the fact that the addition of an appropriate amount of *ZBS1* glass refines the microstructure of ceramics, resulting in an increase in the BDS and enhancing the polarization behavior of the ceramics.

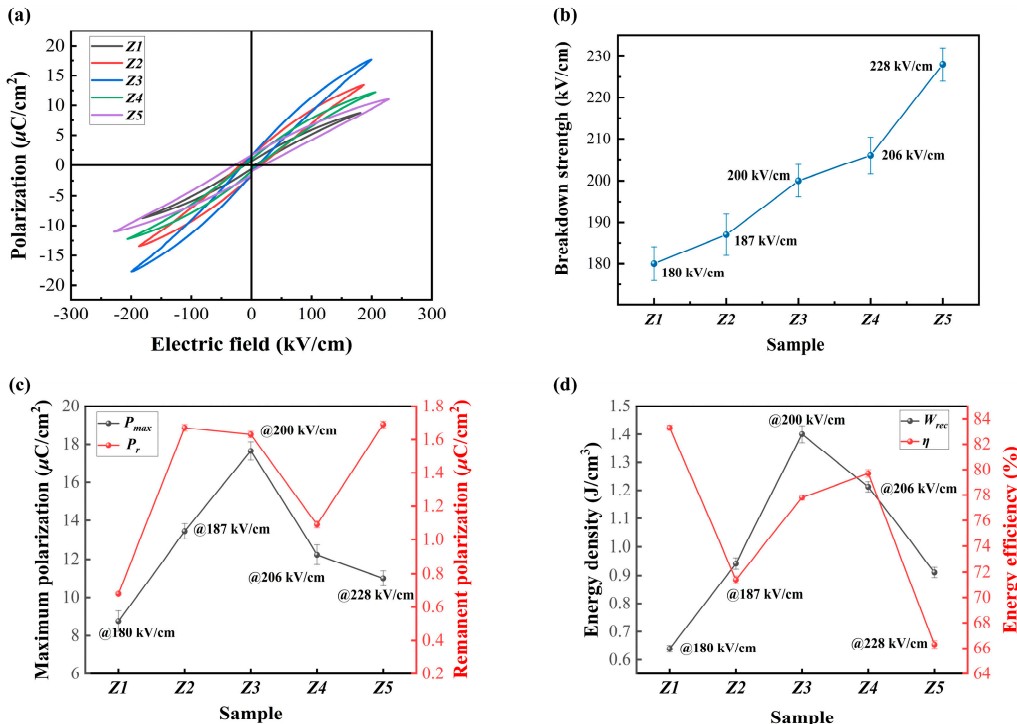

**Figure 6.** (**a**) $P{-}E$ hysteresis loops, (**b**) breakdown strength, (**c**) polarization, and (**d**) energy storage properties of the glass-added BT-BMN ceramics.

## 4. Conclusions

In conclusion, $(100 − x)$BT-BMN–$x$ZBS glass-added ceramics with improved energy storage density and efficiency were fabricated via the solid-state reaction method. The phase structure, microstructure, dielectric properties, and energy storage performance of these ceramic samples were systematically investigated. XRD analysis revealed that the glass-added ceramics exhibited a perovskite structure and the optimal relative density was obtained with *ZBS* of 6 wt%. The average grain size reduced remarkably with the addition of *ZBS*. The samples showed great dielectric temperature stability, improved breakdown strength, and slim *P–E* loops. When $x = 6$, the optimal energy storage performance was observed with a recoverable energy density of 1.39 J/cm$^3$, marking a 23% increase compared to a previous study, and an energy storage efficiency of 78.3%. All of these energy storage properties indicate that these BT-BMN–*ZBS* ceramics can be considered as an ideal candidate for high-pulse-power energy storage devices.

**Author Contributions:** Conceptualization, H.H.; methodology, P.X. and M.X.; formal analysis, P.X. and M.X.; data curation, P.X. and M.X.; writing—original draft preparation, P.X.; writing—review and editing, H.H.; supervision, H.L. and Z.Y. All authors have read and agreed to the published version of the manuscript.

**Funding:** This work was funded by the National Key Research and Development Program of China (No. 2023YFB3812200) and Guangdong Basic and Applied Basic Research Foundation (No. 2022B1515120041).

**Data Availability Statement:** The original contributions presented in the study are included in the article. Further inquiries can be directed to the corresponding author.

**Conflicts of Interest:** The authors declare no conflicts of interest.

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
