# Peer review of "Improving the Energy Storage Performance of Barium Titanate-Based Ceramics through the Addition of ZnO-Bi2O3-SiO2 Glass"

_crystals, doi:10.3390/cryst14030242_

Round 1

Reviewer 1 Report

Comments and Suggestions for Authors

The article is devoted to the study of the properties of lead-free ceramics obtained using the solid-phase synthesis method, which are planned to be used as energy sources. In general, the presented text of the article is quite interesting and presented with good scientific validity, which makes this work one of the most promising studies in this area for the further development of this area. However, before this work can be accepted for publication, the authors should answer a number of questions related to expanding the data obtained and improving the results presented.

1. The observed changes in grain sizes must be compared with the results of structural changes, whether there is a correlation between them, since the presented X-ray diffraction patterns clearly show a change in the shape of the lines, characteristic of a decrease in the size of coherent scattering regions.

2. The authors should provide more details about why the breakdown strength increases with increasing glass concentration while the dielectric constant decreases. Is this related to size factors and if so, what role does dislocation density play in these changes?

3. Did the authors calculate the ratio between the amorphous and crystalline parts of the ceramics under study depending on the variation of the components? If so, then these data must be compared with the results of density depending on the sintering temperature.

4. The change in crystal lattice parameters presented in the table should be explained in terms of the effects of substitution or thermal expansion of the crystal structure upon heating.

5. The choice of temperature range is quite small, in connection with this the question arises of how the temperature was controlled during the synthesis process, since at high temperatures jumps during heating within the range of 10 - 20 ° C are possible.

Reviewer 2 Report

Comments and Suggestions for Authors

1. Line 84 page 2 It is need to correct name - Scanning Electron Microscope.

2. Fig.2 Authors write:  "It can be observed that the sintering temperature at which the maximum bulk density is achieved for Z1 and Z2 samples is 1100°C, whereas for Z3, Z4, and Z5 samples, it decreases to 1080°C".

For such a statement, it is necessary to indicate what the spread of data is. What is the accuracy of the method. This maximum may be an overshoot at line for Z1 and simply a rise for line for Z2. Question - how many measurements were taken for each point on the sintering temperature? You need to plot the “magnitude of scatter” on the graph.

3. Fig. 6 b,c,d it is need to add also “magnitude of scatter”.

Reviewer 3 Report

Comments and Suggestions for Authors

High-performance, lead-free ceramics are vital for powerful energy storage devices. In this study, 0.9BaTiO3-0.1Bi (Mg2/3Nb1/3)O3 (BT-BMN) ceramics were created with ZnO-Bi2O3-SiO2 (ZBS) glass additives. X-ray diffraction showed a perovskite structure, with maximum density at x = 6. As the glass additive increased, grain size decreased, dielectric constant dropped, and breakdown strength increased. Optimal energy storage density (1.39 J/cm3) and efficiency (78.3%) were achieved at x = 6, making these ceramics promising for high-energy storage pulse power devices. If the following problems are well-addressed, this reviewer believes that the essential contribution of this article is vital for ceramic based energy storage devices.

1.      What is the detailed mechanism behind the observed changes in average grain size with the addition of ZBS glass? How do these changes affect the overall performance of ceramics?

2.      How does the performance of BT-BMN – ZBS ceramics vary under different temperature conditions, and can the ceramics maintain their energy storage properties across a wide temperature range?

3.      How well do the BT-BMN – ZBS ceramics integrate into existing or new energy storage device architectures? What are the challenges and opportunities for practical device implementation?

Round 2

Reviewer 1 Report

Comments and Suggestions for Authors

The authors have answered all the questions posed, the article can be accepted for publication.